# Verb Conjugation in Transformers Is Determined by Linear Encodings of Subject Number

**Sophie Hao**
New York University
sophie.hao@nyu.edu

**Tal Linzen**
New York University
linzen@nyu.edu

## Abstract

Deep architectures such as Transformers are sometimes criticized for having uninterpretable "black-box" representations. We use causal intervention analysis to show that, in fact, some linguistic features are represented in a linear, interpretable format. Specifically, we show that BERT's ability to conjugate verbs relies on a linear encoding of subject number that can be manipulated with predictable effects on conjugation accuracy. This encoding is found in the subject position at the first layer and the verb position at the last layer, but is distributed across positions at middle layers, particularly when there are multiple cues to subject number.

## 1 Introduction

Although neural network language models (LMs) are sometimes viewed as uninterpretable "black boxes," substantial progress has been made towards understanding to which linguistic regularities LMs are sensitive and how they represent those regularities, in particular in the case of syntactic constraints such as subject–verb agreement. This progress includes not only the discovery that LM predictions adhere to such constraints (e.g., Linzen et al., 2016), but also the development of tools that have revealed encodings of syntactic features in hidden representations (Adi et al., 2017; Giulianelli et al., 2018, among others).

Most prior work on LMs' internal vector representations has demonstrated the *existence* of syntactic information in those vectors, but has not described how LMs *use* this information. This paper addresses the latter question using a causal intervention paradigm proposed by Ravfogel et al. (2021). We first hypothesize that at least one hidden layer of BERT (Devlin et al., 2019) encodes the grammatical number of third-person subjects and verbs in a low-dimensional *number subspace* of the hidden representation space, where singular number is linearly separable from plural number. We then predict that *intervening* on the hidden space by reflecting hidden vectors to the opposite side of the number subspace will cause BERT to generate plural conjugations for singular subjects, and *vice versa*. Our experiment confirms this prediction dramatically: BERT's verb conjugations are 91% *correct* before the intervention, and up to 85% *incorrect* after the intervention.

In addition to these findings, our experiment makes observations regarding the *location* of subject number encodings across token positions, and how it changes throughout BERT's forward computation. We find that subject number encodings originate in the position of the subject at the embedding layer, and move to the position of the inflected verb at the final layer. When the sentence contains additional cues to subject number beyond the subject itself, such as an embedded verb that agrees with the subject, subject number encodings propagate to other positions of the input at middle layers.

Unlike our study, prior counterfactual intervention studies have not been able to consistently produce the expected changes in LM behavior. In Finlayson et al. (2021) and Ravfogel et al. (2021), for example, interventions only cause slight degradations in performance, leaving LM behavior mostly unchanged. These numerically weaker results show that LM behavior is *influenced* by linear feature encodings, but is ultimately driven by other representations, which may have a non-linear structure. In contrast, our results show that the linear encoding of subject number *determines* BERT's ability to conjugate verbs. The mechanism behind verb conjugation is therefore linear and interpretable, far from being a black box.[1]

## 2 Background and Related Work

This study contributes to a rich literature on the representation of natural language syntax in LMs.

---

[1] Code for our experiment is available at: https://github.com/yidinghao/causal-conjugation

We briefly review this literature in this section; a more comprehensive overview is offered by Lasri et al. (2022).

**LMs and Syntax.** A popular approach to the study of syntax in LMs is through the use of *behavioral* experiments. An influential example is Linzen et al. (2016), who evaluate English LSTM LMs on their ability to conjugate third-person present-tense verbs. Since verb conjugation depends on syntactic structure in theory, this study can be viewed as an indirect evaluation of the LM's knowledge of natural language syntax. Linzen et al.'s methodology for evaluating verb conjugation is to compare probability scores assigned to different verb forms, testing whether an LM is more likely to generate correctly conjugated verbs than incorrectly conjugated verbs. Follow-up studies such as Marvin and Linzen (2018), Warstadt et al. (2019), and Gauthier et al. (2020) have refined the behavioral approach by designing challenge benchmarks with experimental controls on the structure of example texts, which allow for fine-grained evaluations of specific linguistic abilities.

**Probing and LM Representations.** Another approach to syntax in LMs is the use of *probing classifiers* (Adi et al., 2017; Belinkov et al., 2017; Hupkes and Zuidema, 2017; Hupkes et al., 2018). By contrast with behavioral studies, probing studies analyze what information is encoded in LM representations. A typical analysis attempts to train the probing classifier to decode the value of a syntactic feature from hidden vectors generated by an LM. If this is successful, then the study concludes that the hidden space contains an encoding of the relevant information about the syntactic feature. When the probing classifier is linear, the study can additionally conclude that the encoding has a linear structure. An overview of probing results for BERT is provided by Rogers et al. (2020).

**Counterfactual Intervention.** Counterfactual intervention enhances the results of a probing study by determining whether a feature encoding discovered by a linear probe is actually used by the LM, or whether the probe has detected a spurious pattern that does not impact model behavior. Early studies such as Giulianelli et al. (2018), Lakretz et al. (2019), Tucker et al. (2021), Tucker et al. (2022), and Ravfogel et al. (2021) provide evidence that manually manipulating representations of subject number can result in causal effects on LM verb

**Counterfactual Intervention:** Let $\lambda_1, \lambda_2, \ldots, \lambda_k$ be the coordinates of $\boldsymbol{h}^{(l,i)}$ along the number subspace. We modify $\boldsymbol{h}^{(l,i)}$ as shown below. If $\alpha \geq 2$, then the modified vector should encode the opposite subject number. If $\alpha = 1$, then the modified vector should contain no information about subject number.

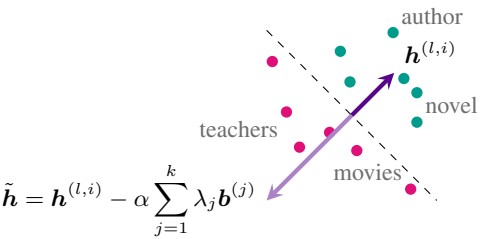

$$\tilde{\boldsymbol{h}} = \boldsymbol{h}^{(l,i)} - \alpha \sum_{j=1}^{k} \lambda_j \boldsymbol{b}^{(j)}$$

**Verb Conjugation:** We predict that intervention with $\alpha \geq 2$ will cause BERT to conjugate verbs incorrectly.

Before Intervention: $\mathbb{P}[\text{is}] > \mathbb{P}[\text{are}]$
After Intervention: $\mathbb{P}[\text{is}] < \mathbb{P}[\text{are}]$

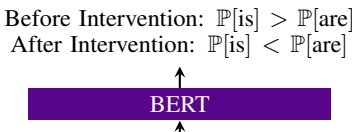

The author that the teachers admire [MASK] happy.

Figure 1: Illustration of our counterfactual intervention (above) and our verb conjugation test (below).

conjugation and other linguistic abilities. The goal of this paper is to present an instance where linear encodings *fully* determine the verb conjugation behavior of an LM.

## 3 Methodology

Let $\boldsymbol{h}^{(l,i)} \in \mathbb{R}^{768}$ be the hidden vector from layer $l$ of BERT$_{\text{BASE}}$ for position $i$. Our hypothesis is that there is an orthonormal basis $\mathbb{B} = \{\boldsymbol{b}^{(1)}, \boldsymbol{b}^{(2)}, \ldots, \boldsymbol{b}^{(768)}\}$ such that for some $k \ll 768$, the first $k$ basis vectors span a *number subspace* that linearly separates hidden vectors for singular-subject sentences from hidden vectors for plural-subject sentences. Our prediction is that the *counterfactual intervention* illustrated in Figure 1, where hidden vectors are reflected to the opposite side of the number subspace, will reverse the subject number encoded in the vectors when applied with sufficient intensity (as determined by the hyperparameter $\alpha$), causing BERT to conjugate the main verb of a sentence as if its subject had the opposite number. This section describes (1) how our counterfactual intervention is defined, (2) how we find the basis vectors for the number subspace, and (3) how we measure the effect of this intervention on verb conjugation.

**Counterfactual Intervention.** Suppose that the hidden vector $h^{(l,i)}$ is computed from an input consisting of a single sentence. The goal of our counterfactual intervention is to transform $h^{(l,i)}$ into a vector $\tilde{h}$ that BERT will interpret as a hidden vector representing the same sentence, but with the opposite subject number. To do so, we first assume that $h^{(l,i)}$ is written in terms of the basis $\mathbb{B}$:

$$h^{(l,i)} = \sum_{j=1}^{768} \lambda_j b^{(j)},$$

where for each $j$, the coordinate $\lambda_j$ is the scalar projection of $h^{(l,i)}$ onto the unit vector $b^{(j)}$:

$$\lambda_j = \left( h^{(l,i)} \right)^\top b^{(j)}.$$

Next, we assume that the coordinates of $h^{(l,i)}$ along the number subspace, $\lambda_1, \lambda_2, \ldots, \lambda_k$, collectively encode the input sentence's subject number, and that $-\lambda_1, -\lambda_2, \ldots, -\lambda_k$ encode the opposite subject number. We compute $\tilde{h}$ by simply moving these coordinates of $h^{(l,i)}$ towards the opposite subject number:

$$\tilde{h} = h^{(l,i)} - \alpha \sum_{j=1}^{k} \lambda_j b^{(j)}.$$

The variable $\alpha$ is a hyperparameter that determines the intensity of the counterfactual intervention. When $\alpha = 1$, the coordinates along the number subspace are set to 0; $\tilde{h}$ is then interpreted as a vector that encodes no information about subject number. If our hypothesis about the number subspace is correct, then counterfactual intervention with $\alpha \geq 2$ should result in a vector $\tilde{h}$ that encodes the opposite subject number.

**Finding the Number Subspace.** We use the *iterative nullspace projection* (INLP, Ravfogel et al., 2020; Dufter and Schütze, 2019) method to calculate the basis for the number subspace. We begin by using BERT to encode a collection of sentences and extracting the hidden vectors $h^{(l,i)}$ in the positions of main subjects. We then train a linear probe to detect whether these hidden vectors came from a singular subject or a plural subject, and take $b^{(1)}$ to be the probe's weight vector, normalized to unit length. To obtain $b^{(j)}$ for $j > 1$, we use the same procedure, but preprocess the data by applying counterfactual intervention with $\alpha = 1$ and $k = j-1$. This erases the subject number information captured by previously calculated basis vectors, ensuring that $b^{(j)}$ is orthogonal to $b^{(1)}, b^{(2)}, \ldots, b^{(j-1)}$.

**Measuring the Effect of Intervention.** We evaluate BERT's verb conjugation abilities using a paradigm based on Goldberg (2019), where masked language modeling is performed on sentences with a third-person subject where the main verb, *is* or *are*, is masked out. We calculate *conjugation accuracy* by interpreting BERT's output as a binary classification, where the predicted label is "singular" if $\mathbb{P}[\text{is}] > \mathbb{P}[\text{are}]$ and "plural" otherwise. To test our prediction about the causal effect of number encoding on verb conjugation, we measure conjugation accuracy before and after intervention with $\alpha \geq 2$. If intervention causes conjugation accuracy to drop from $\approx 100\%$ to $\approx 0\%$, then we conclude that we have successfully encoded the opposite subject number into the hidden vectors. If conjugation accuracy drops to $\approx 50\%$, then number information has been erased, but we cannot conclude that the opposite subject number has been encoded.

## 4 Experiment

We test our prediction by performing an experiment using the `bert-base-uncased` instance of BERT. For each layer, we apply counterfactual intervention and measure its effect on conjugation accuracy. We perform two versions of our experiment: one where intervention is applied to all hidden vectors ("global intervention"), and one where intervention is only applied to hidden vectors in the subject position ("local intervention"). We repeat our experiment five times, with each trial using linear probes trained on a freshly sampled, balanced dataset of 4,000 hidden vectors.

**Data.** We use data from Ravfogel et al. (2021), which consist of sentences with a relative clause intervening between the main subject and the main verb (e.g., *The **author** that the teacher admires **is** happy*). We sample the INLP training vectors from their training split, and we use their testing split to measure conjugation accuracy.

**Hyperparameters.** We tune the hyperparameters $\alpha$ (intensity of intervention) and $k$ (dimensionality of the number subspace) using a grid search over the range $\alpha \in \{2, 3, 5\}$ and $k \in \{2, 4, 8\}$.

**Main Results.** Figure 2 shows our results. The values of $\alpha$ and $k$ do not affect our results qualitatively, but they do exhibit a direct relationship with the magnitude of the effect of intervention on conjugation accuracy. We focus on the results for

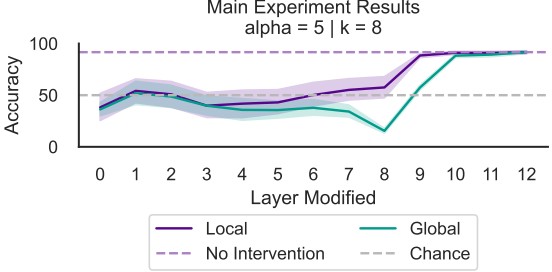

Figure 2: The effect of local and global intervention on conjugation accuracy. Error bands represent 95% confidence intervals obtained from 5 samplings of INLP training vectors.

**Redundant Cues:**
The **author** that **admires** the teachers **is** happy.

**No Redundant Cues:**
The **author** that the teachers admire **is** happy.

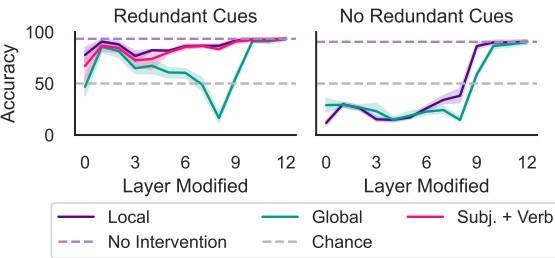

Figure 3: The linear encoding of subject number spreads to positions other than the subject when there are redundant cues to subject number, such as an **embedded verb**. In the "subj. + verb" condition, intervention is applied to the subject and embedded verb positions. Error bands represent 95% confidence intervals obtained from 5 samplings of INLP training vectors.

$\alpha = 5$ and $k = 8$, which exhibit the greatest impact of intervention on conjugation accuracy. The full hyperparameter tuning results can be found in Appendix A.

Our prediction is confirmed when global intervention, where hidden vectors across all positions are modified, is applied to layer 8. Verb conjugations are 91.7% *correct* before intervention, but 84.6% *incorrect* after intervention. Local intervention on layer 8, where only the hidden vector in the subject position is modified, has a much weaker effect, only causing conjugation accuracy to drop to 57.5% (42.5% incorrect). These results show that BERT indeed uses a linear encoding of subject number to comply with subject–verb agreement. The location of this linear encoding is not confined to the position of the subject, but is rather distributed across multiple positions.

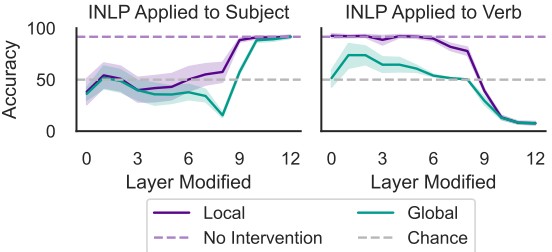

Figure 4: Conjugation accuracy drops to 7.6% when intervening on layer 12 with INLP training vectors extracted from the verb position (right) instead of the subject position (left). Error bands represent 95% confidence intervals obtained from 5 samplings of INLP training vectors.

**Redundant Cues to Number.** Some sentences in our training and testing data contain an embedded verb that agrees with the main subject. For example, in the sentence *The **author** that **admires** the teacher is happy*, the singular verb ***admires*** agrees with the subject ***author***. Since we can deduce the number of the subject from the number of this embedded verb, even in the absence of any direct access to a representation of the subject, in these sentences the embedded verb serves as a *redundant cue* to subject number.

Figure 3 shows the effects of intervention broken down by the presence of cue redundancy. When there is no redundancy, near-zero conjugation accuracy is observed after both local and global intervention applied to layers 0–6. This shows that when the subject is the only word that conveys subject number, verb conjugation depends solely on the hidden vector in the subject position. By contrast, local intervention has no effect on conjugation accuracy in the presence of redundant cues, and neither does intervention in the positions of the subject and the embedded verb (the "subj. + verb" condition). This shows that the presence of a redundant cue to subject number causes BERT to distribute the encoding of subject number to multiple positions.

**Upper Layers.** In layers 10–12, neither local nor global intervention has any effect on conjugation accuracy. We hypothesize that this is because, at these layers, the INLP linear probe cannot identify the number subspace using training vectors extracted from the subject position of sentences. To test this hypothesis, we extract INLP training vectors from the position of the main verb instead of the subject as before, and apply local interven-

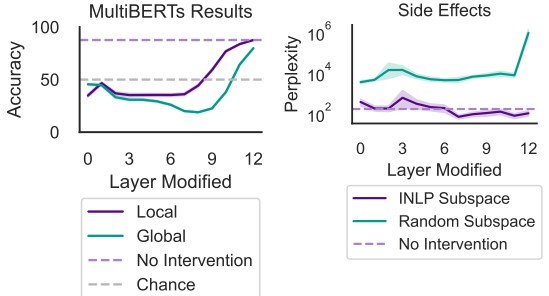

Figure 5: Left: Similar results are obtained when repeating the experiment using the MultiBERT models. Right: Intervention on number-neutral words has no adverse effect on perplexity. Error bands represent 95% confidence intervals obtained from 5 samplings of INLP training vectors.

tion to the position of the masked-out main verb. Supporting our hypothesis, both local and global intervention result in near-zero conjugation accuracy (Figure 4, right), showing that at upper layers, only the position of the main verb is used by BERT for conjugation.

**Robustness.** To verify that our results are robust to differences in model instance, we repeat our experiment using the MultiBERTs (Sellam et al., 2022), a collection of 25 BERT$_{BASE}$ models pretrained from different random initializations. As shown in the left side of Figure 5, we obtain similar results to Figure 2, indicating that our findings are not specific to `bert-base-uncased`.

**Side Effects.** Does the number subspace encode information *beyond* number? To test this, we apply intervention to *number-neutral words* (i.e., all words other than nouns and verbs) along the number subspace. We find that this has no effect on masked language modeling perplexity for those words (Figure 5). In contrast, intervention on number-neutral words along a random 8-dimensional representation subspace increases perplexity by a factor of 52.8 on average. This shows that the number space *selectively* encodes number, such that manipulating hidden vectors along the number subspace does not affect predictions unrelated to number.

## 5  Discussion

In this section, we discuss our results in relation to our current knowledge about linear representations.

**BERT Layers.** Probing studies have found that lower layers of BERT encode lexical features, while middle layers encode high-level syntactic features and upper layers encode task-specific features (Hewitt and Manning, 2019; Jawahar et al., 2019; Kovaleva et al., 2019; Liu et al., 2019; Tenney et al., 2019). Our results confirm this in the case of cue redundancy: at layer 8, the representation of subject number is not tied to any position; while at layer 12, it is tied to the [MASK] position, where it is most relevant for masked language modeling. When there is no cue redundancy, however, subject number is tied to the subject position until layer 9, suggesting that subject number is treated as a lexical feature of the subject rather than a sentence-level syntactic feature.

**Effect Size.** Prior counterfactual intervention studies only report marginal changes in performance after intervention (e.g., Kim et al., 2018; Dalvi et al., 2019; Lakretz et al., 2019; Finlayson et al., 2021; Ravfogel et al., 2021). For example, the largest effect size reported by Ravfogel et al. (2021) is no more than 35 percentage points. These results suggest that the linear encoding is only a relatively small part of the model's representation of the feature. Our results improve upon prior work by identifying an aspect of LM behavior that is entirely driven by linear feature encodings.

## 6  Conclusion

Using a causal intervention analysis, this paper has revealed strong evidence that BERT hidden representations contain a linear encoding of main subject number that is used for verb conjugation during masked language modeling. This encoding originates from the word embeddings of the main subject and possible redundant cues, propagates to other positions at the middle layers, and migrates to the position of the masked-out verb at the upper layers. The structure of this encoding is interpretable, such that manipulating hidden vectors along this encoding results in predictable effects on conjugation accuracy.

Our clean and interpretable results offer subject number as an example of a feature that a large language model might encode using a straightforwardly linear-structured representation scheme. For future work, we pose the question of what kinds of features may admit similarly strong results from a causal intervention study like this one.

## Limitations

Below we identify possible limitations of our approach.

**Experimental Control.** By utilizing Ravfogel et al.'s (2021) dataset, where sentences adhere to a uniform syntactic template, we have exerted tight experimental control over the structure of our test examples. This control has allowed us, for instance, to identify the qualitatively distinct results from Figure 3 between inputs with and without a redundant cue to subject number. In a more naturalistic setting, it is possible that verb conjugation may be conditioned by factors other than a linear encoding of subject number, such as semantic collocations or discourse context.

**Asymmetry of Findings.** Although we have shown that BERT uses a linear encoding of subject number to conjugate verbs, we can never prove using our approach that BERT *does not* use a linear encoding of a feature to some end. In the instances where we are unable to encode the opposite subject number, we cannot rule out the possibility that BERT uses a linear encoding of subject number that cannot be detected using INLP.

## Ethical Considerations

We do not foresee any ethical concerns arising from our work.

## Acknowledgments

We thank the EMNLP 2023 reviewers and area chairs as well as members of the New York University (NYU) Computation and Psycholinguistics Lab for their feedback on this paper.

This work was supported in part through the NYU IT High Performance Computing resources, services, and staff expertise. This material is based upon work supported by the National Science Foundation (NSF) under Grant No. BCS-2114505.

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

## A    Hyperparameter Tuning Results

Our full hyperparameter tuning results are shown in Figure 6.

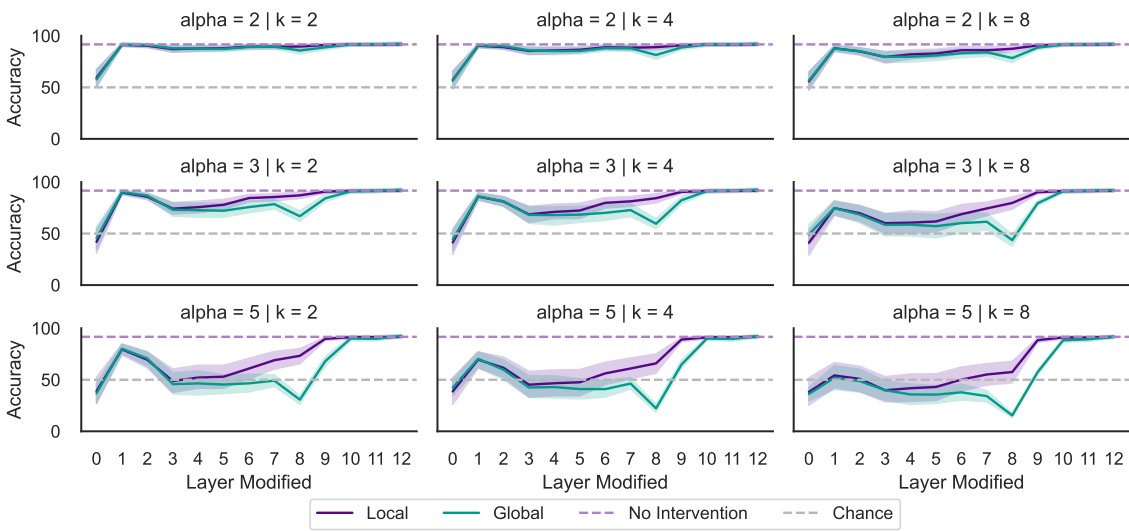

Figure 6: Hyperparameter tuning results for $\alpha$ (intensity of counterfactual intervention) $k$ (dimensionality of the number subspace). Error bands represent 95% confidence intervals obtained from 5 samplings of INLP training vectors.