# OpenReview forum: "Verb Conjugation in Transformers Is Determined by Linear Encodings of Subject Number"
_EMNLP/2023/Conference — EMNLP 2023 Findings_

### Official Review · Reviewer_vo72 · 2023-07-27

**Soundness:** 3

**Excitement:**

3: Ambivalent: It has merits (e.g., it reports state-of-the-art results, the idea is nice), but there are key weaknesses (e.g., it describes incremental work), and it can significantly benefit from another round of revision. However, I won't object to accepting it if my co-reviewers champion it.

**Paper Topic And Main Contributions:**

This paper presents an extension of the "BERTology". By probing the vectors from different layers of the BERT model using the causal internvention analysis, this paper shows that the BERT model contains a linear encoding of the representations for the task of verb conjugation.

**Reasons To Accept:**

- It's a well-written paper. The experiment setup is neat and the result analysis is clear.
- This paper proposes an interesting finding that the BERT model has linear encoded representation for the verb conjugation task. They show strong evidences and robust analysis to support the finding under their specific experiment setup.
- The paper might be able to inspire future work to investigate other linguistic features embedded in the BERT using the similar paradigm

**Reasons To Reject:**

This paper focuses on examining whether the transformer models represent some linguistic features linearly by applying the causal intervention analysis. They did the analysis on a specific setup, e.g, a single model type. I think the analysis can be better supported and more useful to the future work by having more variances added to the experiment setup:
- BERT models come with difference sizes (base, large, etc.)
- other up-to-date transformer models or LLMs
- texts with difference syntactic / semantic structures (authors were aware of this in the limitation section)

**Reproducibility:**

4: Could mostly reproduce the results, but there may be some variation because of sample variance or minor variations in their interpretation of the protocol or method.

**Reviewer Confidence:**

4: Quite sure. I tried to check the important points carefully. It's unlikely, though conceivable, that I missed something that should affect my ratings.

**Typos Grammar Style And Presentation Improvements:**

This is not a weakness so I put it here. The title of this paper I think is kind of "exaggerating" in a way that it's so general that readers might not be able to get a gist on what specific linguistic feature is being discussed in the paper. There is no harm I think putting words relating to verb conjugation in the title.

---

> ### Author Rebuttal · Authors · 2023-08-28
>
> Thank you for your review.
>
> The points about testing a wider range of model types is fair and well-taken. We would like to mention that we do consider a number of different syntactic structures in our testing data, even if those structures are subject to experimental control, as we describe in our Limitations section. And while we have only considered BERT-base models, we have repeated our experiments on a number of other BERT-base models (i.e., the MultiBERTs) trained from different random initializations.
>
> As for the title, we are happy to change it in order to better convey the content of the paper, perhaps to: “Verb Conjugation in Transformers is Determined by Linear Encodings of Subject Number.”

---

### Official Review · Reviewer_SDLf · 2023-08-04

**Soundness:** 4

**Excitement:**

3: Ambivalent: It has merits (e.g., it reports state-of-the-art results, the idea is nice), but there are key weaknesses (e.g., it describes incremental work), and it can significantly benefit from another round of revision. However, I won't object to accepting it if my co-reviewers champion it.

**Missing References:**

A short discussion of the notion of "encoding" in the sense of what model representations encode vs. what the models actually uses for predictions would be in order. https://arxiv.org/pdf/2204.08831.pdf provides an overview of relevant literature.

**Paper Topic And Main Contributions:**

The paper argues that there is a subpace in the vector space of BERT token representations that encodes grammatical number, i.e. the information necessary for correctly predicting the form of the verb as regards the correct number agreement with the subject. The authors show that it is possible to find an orthonormal lower-dimensional basis for this subspace and then use the elements of this basis to erase or invert the information about grammatical number contained in token representations. The authors use iterative nullspace projection by Ravfogel et al. (2020) to compute the basis and measure the effect of their interventions using the dataset on number agreement compiled by Goldberg (2019). They show that their method is largely robust to the choice of hyperparameters (dimensionality of the subspace and the "intensity of intervention", essentially a general weight for the sum of weighted basis vectors), but that it is important to carefully choose the target layer and token depending on the structure of the sentence. The authors repeat their experiments on the MultiBERTs showing that their results are not limited to the uncased version of BERT base.

**Reasons To Accept:**

The paper provides a clear link between a well-defined manipulation of the vector space of token embeddings and an important linguistic property. It focuses on a single language and model type, but inside these constraints it does hyperparameter checking and repeats the experiment with different models from the same class (BERT base uncased).

**Reasons To Reject:**

Even though the object of study and the results are clearly presented, the actual contributions of the paper are a bit muddled. The fact that "Transformers Represent Some Linguistic Features Linearly" is not, strictly speaking, a new finding: the whole line of work on iterative nullspace projection is predicated on this idea, as it tries to erase/manipulate linearly seaparable properties inside Transformer representations.

The authors say that previous works "report effect sizes smaller than 50 percentage points... [and] do not show that the opposite value of the feature can be encoded" (lines 69-73). This is true, but some of the previous work seems to have been solving a slightly different task using a very similar method, and in the end it remains unclear if (i) we are presented with numerically better results that are due to an improved methodology or (ii) the authors claim that an improved methodology coupled with a change in the task definition leads to new knowledge about Transformers in general, i.e. that they encode some linguistic features _almost exlusively_ linearly.

Moreover, "encoding" here is equated with "represented in such a way that their effect on the behaviour of the model can be undone using linear methods". As we know, Transformers can also encode information that they do not use for predictions and that can be still recovered with non-linear classifiers.

**Reproducibility:**

3: Could reproduce the results with some difficulty. The settings of parameters are underspecified or subjectively determined; the training/evaluation data are not widely available.

**Reviewer Confidence:**

3: Pretty sure, but there's a chance I missed something. Although I have a good feel for this area in general, I did not carefully check the paper's details, e.g., the math, experimental design, or novelty.

**Typos Grammar Style And Presentation Improvements:**

The presentation of the methodology is a bit surprising: the key formula is presented inside a figure and its elements are not discussed at all in the main text. Nothing is said at all about how to calculate $\lambda$'s, which is probably recoverable but should be more explicit.

"Some" in the title is replaced with a single feature, grammatical number, already in the abstract. Unless the authors want to add number to other features discussed in previous works, with arguably weaker results, this looks like overselling.

---

> ### Author Rebuttal · Authors · 2023-08-28
>
> Thank you for your review.
>
> We do in fact believe that our paper presents new knowledge about Transformers—namely, as you describe, that some high-level syntactic features are encoded _almost exclusively linearly_ by Transformers, particularly in the middle layers. While we agree that the existence of linear encodings in general is not a new finding, our paper does present novel findings on:
> - which features have a linear encoding,
> - where (across layers and positions) this linear encoding is found, and
> - how important this linear encoding is compared to other drivers of model behavior.
>
> Most importantly, whereas previous papers have shown that linear encodings _influence_ model behavior, our paper is the first to present a syntactic feature whose linear encoding _determines_ model behavior. Crucially, the latter conclusion is only possible if the effect size of intervention exceeds 50 percentage points and approaches 100 percentage points, which is what our paper shows. Therefore, the contribution of our paper is not merely that we obtain better numerical results; it's that these improved results allow us to reach a conclusion that previous papers have not been able to reach.
>
> In order to make these arguments more clear, we are happy to change the title to: “Verb Conjugation in Transformers is Determined by Linear Encodings of Subject Number.”
>
> We are happy to incorporate a discussion of linear vs. non-linear encodings, as well as the distinctions between diagnostic, behavioral, and causal probing (in the terminology of Lasri et al., 2023, linked in the review). As for the calculation of $\lambda$s, they are simply the scalar projections of the hidden vectors onto the (unit-length) basis vectors, given by the following formula:
> $$\lambda_j = \left( \boldsymbol{h}^{(l, i)} \right)^\top \boldsymbol{b}^{(j)}$$
> This is a standard formula in linear algebra, and we are happy to explain this explicitly in the final version if this paper is accepted.

---

### Official Review · Reviewer_cZfP · 2023-08-04

**Typos Grammar Style And Presentation Improvements:** N/A
**Soundness:** 4

**Excitement:**

3: Ambivalent: It has merits (e.g., it reports state-of-the-art results, the idea is nice), but there are key weaknesses (e.g., it describes incremental work), and it can significantly benefit from another round of revision. However, I won't object to accepting it if my co-reviewers champion it.

**Missing References:**

N/A

**Paper Topic And Main Contributions:**

This paper investigates whether the subject number is linearly encoded in BERT's hidden states, and shows that the representation can be manipulated to encode the opposite number information with linear transformations. A great number of experiments are conducted to analyze the effect of different clues, the robustness and side effects.

**Questions For The Authors:**

Question A: Will subjects happen to have sub-tokens? If so, how do you deal with it when extracting corresponding hidden vectors?


**Reasons To Accept:**

The paper is well-written and clearly structured.

This paper has conducted some interesting experiments to analyse different aspects, such as the robustness, side effects.

**Reasons To Reject:**

The study is focused on one type of model and one type of linguistic feature. The title of the paper would be better supported with experiments in more settings (e.g., different transformer-based models).

Given previous work like Finlayson et al. (2021) and Ravfogel et al. (2021), I do not think this paper is novel enough or provides novel enough conclusions to publish at EMNLP. The methods are not new and the results are not surprising.

**Reproducibility:**

4: Could mostly reproduce the results, but there may be some variation because of sample variance or minor variations in their interpretation of the protocol or method.

**Reviewer Confidence:**

3: Pretty sure, but there's a chance I missed something. Although I have a good feel for this area in general, I did not carefully check the paper's details, e.g., the math, experimental design, or novelty.

---

> ### Author Rebuttal · Authors · 2023-08-28
>
> Thank you for your review. We would like to discuss the novelty of our paper, and how our results relate to previous ones.
>
> The novelty of our paper is that our conclusions are much more justified than in previous papers. While prior studies have found _some_ evidence in favor of linearly encoded syntactic features, our paper is the first to provide _conclusive_ evidence for it.
>
> In Ravfogel et al. (2021), for example, causal intervention only had a slight impact on the model’s behavior. While this does show that their target feature (membership in a relative clause) _has_ a linear encoding, the weakness of their effect size also shows that this linear encoding is not particularly important for the model. On the other hand, our experiments show that linear encodings _determine_ model behavior in the case of verb conjugation: manipulating the linear encoding predictably alters model behavior almost 100% of the time.
>
> This finding is novel, and we argue that it is in fact surprising. In past work we have only seen linear encodings that _influence_ model behavior; our paper is the first to present a linear encoding that _determines_ model behavior. We believe it is quite surprising that the linear encoding of subject number can override all other variables that might drive model behavior in the case of verb conjugation.
>
> In addition to this finding, we also expand upon previous work by pinpointing the location of our linear encodings, both across layers of the Transformer and across positions of the input. This kind of analysis has not been previously done in causal intervention studies, and we believe our results do provide a novel understanding of Transformer representations.
>
> As for Question A: the training data has exactly one subject that spans two tokens, and the testing data only has single-token subjects. In the case of this one subject, we treat both tokens as “subjects,” in the sense that hidden representations for both positions are included as possible SVM training vectors. Because this particular subject is rare, and because only 2,000 randomly sampled vectors are included in the SVM training data, we do not believe that this substantially affects our results.

---

### Meta-Review · Area_Chair_qXxm · 2023-09-13

**Recommendation:** 3

**Metareview:**

The paper argues that there is a subpace in the vector space of BERT token representations that encodes grammatical number, i.e. the information necessary for correctly predicting the form of the verb as regards the correct number agreement with the subject. The authors show that it is possible to find an orthonormal lower-dimensional basis for this subspace and then use the elements of this basis to erase or invert the information about grammatical number contained in token representations. The authors use iterative nullspace projection by Ravfogel et al. (2020) to compute the basis and measure the effect of their interventions using the dataset on number agreement compiled by Goldberg (2019). They show that their method is largely robust to the choice of hyperparameters (dimensionality of the subspace and the "intensity of intervention", essentially a general weight for the sum of weighted basis vectors), but that it is important to carefully choose the target layer and token depending on the structure of the sentence. The authors repeat their experiments on the MultiBERTs showing that their results are not limited to the uncased version of BERT base.

While the paper is interesting and well written it seems that the novelty and experimental setup are somewhat limited (i.e. the excitement is somewhat limited).

---

### Decision · Program_Chairs · 2023-10-07

**Decision:**

Accept-Findings

**Comment:**

The paper argues that there is a subpace in the vector space of BERT token representations that encodes grammatical number, i.e. the information necessary for correctly predicting the form of the verb as regards the correct number agreement with the subject. The authors show that it is possible to find an orthonormal lower-dimensional basis for this subspace and then use the elements of this basis to erase or invert the information about grammatical number contained in token representations. The authors use iterative nullspace projection by Ravfogel et al. (2020) to compute the basis and measure the effect of their interventions using the dataset on number agreement compiled by Goldberg (2019). They show that their method is largely robust to the choice of hyperparameters (dimensionality of the subspace and the "intensity of intervention", essentially a general weight for the sum of weighted basis vectors), but that it is important to carefully choose the target layer and token depending on the structure of the sentence. The authors repeat their experiments on the MultiBERTs showing that their results are not limited to the uncased version of BERT base.

While the paper is interesting and well written it seems that the novelty and experimental setup are somewhat limited (i.e. the excitement is somewhat limited).